# New Non-Invasive Imaging Technologies in Cardiac Transplant Follow-Up: Acquired Evidence and Future Options

**DOI:** 10.3390/diagnostics13172818

**Published:** 2023-08-31

**Authors:** Valeria Pergola, Giulia Mattesi, Elena Cozza, Nicola Pradegan, Chiara Tessari, Carlo Maria Dellino, Maria Teresa Savo, Filippo Amato, Annagrazia Cecere, Martina Perazzolo Marra, Francesco Tona, Andrea Igoren Guaricci, Giorgio De Conti, Gino Gerosa, Sabino Iliceto, Raffaella Motta

**Affiliations:** 1Cardiology Unit, Department of Cardiac, Thoracic and Vascular Sciences and Public Health, University of Padua, 35128 Padua, Italy; giulia.mattesi@aopd.veneto.it (G.M.); lillo.dellino79@gmail.com (C.M.D.); mariateresa.savo@studenti.unipd.it (M.T.S.); filippo.amato@studenti.unipd.it (F.A.); annagrazia.cecere@unipd.it (A.C.); martina.perazzolomarra@unipd.it (M.P.M.); francesco.tona@unipd.it (F.T.); sabino.iliceto@unipd.it (S.I.); 2Cardiac Surgery Unit, Department of Cardiac, Thoracic, Vascular Sciences and Public Health, University of Padua, 35122 Padua, Italy; nicola.pradegan@aopd.veneto.it (N.P.); chiara.tessari@aopd.veneto.it (C.T.); gino.gerosa@unipd.it (G.G.); 3Department of Emergency and Organ Transplantation, Institute of Cardiovascular Disease, University Hospital “Policlinico” of Bari, 70124 Bari, Italy; andreaigoren.guaricci@uniba.it; 4Radiology Unit, Padova University Hospital, 35128 Padua, Italy; giorgio.deconti@aopd.veneto.it; 5Unit of Radiology, Department of Medicine, Medical School, University of Padua, 35122 Padua, Italy; raffaella.motta@unipd.it

**Keywords:** heart transplantation, cardiac allograft vasculopathy, atrial function, strain echocardiography, stress echocardiography, coronary computed tomography angiography, cardiac magnetic resonance

## Abstract

Heart transplantation (HT) is the established treatment for end-stage heart failure, significantly enhancing patients’ survival and quality of life. To ensure optimal outcomes, the routine monitoring of HT recipients is paramount. While existing guidelines offer guidance on a blend of invasive and non-invasive imaging techniques, certain aspects such as the timing of echocardiographic assessments and the role of echocardiography or cardiac magnetic resonance (CMR) as alternatives to serial endomyocardial biopsies (EMBs) for rejection monitoring are not specifically outlined in the guidelines. Furthermore, invasive coronary angiography (ICA) is still recommended as the gold-standard procedure, usually performed one year after surgery and every two years thereafter. This review focuses on recent advancements in non-invasive and contrast-saving imaging techniques that have been investigated for HT patients. The aim of the manuscript is to identify imaging modalities that may potentially replace or reduce the need for invasive procedures such as ICA and EMB, considering their respective advantages and disadvantages. We emphasize the transformative potential of non-invasive techniques in elevating patient care. Advanced echocardiography techniques, including strain imaging and tissue Doppler imaging, offer enhanced insights into cardiac function, while CMR, through its multi-parametric mapping techniques, such as T1 and T2 mapping, allows for the non-invasive assessment of inflammation and tissue characterization. Cardiac computed tomography (CCT), particularly with its ability to evaluate coronary artery disease and assess graft vasculopathy, emerges as an integral tool in the follow-up of HT patients. Recent studies have highlighted the potential of nuclear myocardial perfusion imaging, including myocardial blood flow quantification, as a non-invasive method for diagnosing and prognosticating CAV. These advanced imaging approaches hold promise in mitigating the need for invasive procedures like ICA and EMB when evaluating the benefits and limitations of each modality.

## 1. Introduction

Heart transplantation (HT) is the primary treatment option for patients with end-stage heart failure, providing significant improvements in survival and quality of life [1]. However, HT patients are at risk of developing various complications during their follow-up. Common complications include early allograft failure, acute graft rejection (AGR), coronary allograft vasculopathy (CAV), renal failure, infections, and cancer [1,2]. Therefore, it is crucial to establish an effective follow-up protocol for HT patients right from the early post-transplant stages.

Current guidelines suggest several examinations to monitor HT patients. Invasive procedures such as invasive coronary angiography (ICA) and endomyocardial biopsy (EBM) are considered the gold standard for evaluating CAV and AGR, respectively [2]. However, the role of non-invasive methods like echocardiography, stress echocardiography, coronary computed tomography angiography (CCTA), positron emission tomography (PET), and cardiac magnetic resonance (CMR) in monitoring HT patients is not well established.

The guidelines do not provide specific recommendations regarding the timing of echocardiographic evaluations, and CMR is not endorsed as an alternative to serial EMBs for rejection monitoring. It should be noted that no alternative imaging-based strategy or biomarkers have been backed as substitutes for EMB in graft rejection monitoring. According to guidelines, ICA remains the gold standard to exclude CAV, typically performed one year after surgery and subsequently every other year [2,3].

Moreover, it is important to recognize that invasive procedures like ICA and EMB carry the risk of complications, including contrast-related kidney injury and procedure-related lesions. Therefore, there is a need to explore non-invasive approaches that can provide valuable information while minimizing the potential risks associated with invasive procedures [4]. Further research and clinical evidence are required to establish the role of non-invasive imaging modalities and biomarkers in the long-term follow-up of HT patients, ensuring a comprehensive and safe monitoring strategy [3,4]. Therefore, in the present paper, we will discuss all the advancements in non-invasive and contrast-saving imaging techniques that have been proposed and investigated over the past few years to identify those that may have the potential to gate invasive procedures, weighing their advantages and disadvantages.

## 2. Echocardiographic Follow-Up in Transplant Recipients

The International Society for Heart and Lung Transplantation (ISHLT) Guidelines for Heart Transplant Recipients do not recommend echocardiography as a primary method for rejection monitoring due to certain limitations specific to HT patients [5]. One major limitation is the increased variability in echocardiographic parameters in HT patients compared with the general population. This variability makes it challenging to establish definitive “normal” parameters for transplanted hearts and to determine cut-off values for detecting allograft rejection [6,7].

In the absence of established cut-off values, the comparison of echocardiographic parameters from follow-up examinations with previous measurements becomes more valuable than relying on absolute values. This approach allows for the detection of changes over time, which may indicate potential graft rejection (GR) [7].

The European Society of Cardiology (ESC) recommendations suggest a comprehensive echocardiographic study as a baseline evaluation six months after cardiac transplantation [3]. Subsequent follow-up studies should be compared to the data obtained from the six-month study. Although there is no single systolic or diastolic parameter that can reliably diagnose GR, a follow-up echocardiographic study that shows no substantial changes from the baseline has a high negative predictive value for rejection [4]. When a deterioration is observed in a specific parameter, it is important to carefully review previous studies to assess the pattern of change over time. Additionally, the presence of multiple abnormalities detected on echocardiography increases the likelihood of GR [4].

While echocardiography has limitations in detecting GR, it still plays a role in the comprehensive evaluation of HT patients. By carefully assessing changes over time and considering multiple abnormalities, echocardiography can contribute to the overall monitoring and management of HT recipients.

In HT recipients, the typical echocardiographic assessment procedure includes both two-dimensional (2D) imaging and spectral/color Doppler imaging. During these examinations, it is essential to determine the dimensions of the four cardiac chambers and major blood vessels, evaluate the functioning of the left ventricle (LV) and right ventricle (RV), examine the performance of heart valves, measure the systolic pulmonary artery pressure (sPAP), and describe the condition of the pericardium. If there are any irregularities or issues with the graft’s structure or functioning, supplementary views and data acquisitions are necessary [3].

LV ejection fraction (EF), which is typically within the normal range after HT, is not an early indicator of graft dysfunction and does not necessarily correlate with the severity of rejection detected with EMB [8]. Additionally, LV EF is not a sensitive marker for AGR [9].

In contrast, changes in LV diastolic function are more sensitive in detecting AGR compared with the reduction in LV EF. Diastolic function in HT patients follows a bimodal pattern [10]. Despite otherwise successful surgeries, HT recipients often experience a decrease in functional capacity [11]. Diastolic dysfunction is a common occurrence in the early stages of transplantation due to factors such as hypervolemia, disparity in heart and body size between the donor and recipient, the impact of organ ischemia, and early rejection [12]. During the initial month after HT, fluid accumulation tends to occur due to systemic inflammatory response and the administration of high doses of corticosteroids. Therefore, elevated filling pressures during this period may be indicative of elevated volume status [10]. Subsequently, diastolic dysfunction is primarily attributed to episodes of rejection, hypertension, and myocardial ischemia caused by CAV [10].

The evaluation of diastolic function has extensively been explored using Doppler indices of mitral inflow. However, the assessment of LV filling is influenced by various factors, including preload conditions, atrial dynamics, morphology (such as dissociation between recipient and donor atrial contractions), LV compliance and contractility, end-systolic volume, and heart rate. Consequently, impaired diastolic function can stem from different causes and is not specific to rejection episodes. Additionally, the elevated heart rate commonly observed in denervated hearts further complicates the assessment of diastolic function, as it frequently leads to the merging of E and A waves [3].

Several studies have investigated the potential of tissue Doppler imaging (TDI) parameters in predicting AGR. Constant TDI velocities, such as a 10% change in e’ compared with baseline, and high TDI velocities (e.g., e′ > 16 cm/s) appear to have good accuracy in excluding AGR rather than detecting it, with a negative predictive value of 92%. However, these parameters still require further validation [3].

To reliably determine elevated pulmonary capillary wedge pressure (PCWP), it is necessary to observe positive results in at least three out of the five parameters: E/A ratio, deceleration time (DT), isovolumic relaxation time (IVRT), E/e′ lateral ratio, and sPAP. While the individual sensitivity of these parameters is relatively low, the probability of elevated PCWP is unlikely if none of the five parameters meet the defined cut-off values [10].

The myocardial performance index (MPI) has been suggested as an early indicator of rejection in HT patients, considering that rejection affects both diastolic and systolic function simultaneously. However, the accuracy of MPI in detecting AGR remains a topic of debate. In a study conducted by Tona et al. [13], the role of the MPI as a marker for long-term allograft dysfunction was evaluated in 154 patients. The findings revealed a gradual increase in the MPI over the course of long-term follow-up in HT patients with preserved LV systolic function. Higher MPI values were observed in patients with multiple episodes of rejection, although no correlation was found with the development of CAV.

## 3. Echocardiography: What New Technologies Add

Strain echocardiography has higher sensitivity and accuracy compared with conventional echocardiography alone [14,15,16]. This becomes particularly important in the early stages of acute cellular rejection, where standard echocardiography may not detect certain pathological changes, such as pericardial effusion, wall thickening, or increased LV mass [14]. However, strain echocardiography can identify features like myocardial edema or fibrosis, which often affect the sub-endocardial muscle fibers and lead to a decline in longitudinal graft function [15,16].

In the context of HT, reduced LV global longitudinal strain (GLS) shortly after transplantation or the absence of improvement in LV GLS between 2 weeks and 3 months post-transplantation has been linked to poor outcomes [17,18]. Therefore, evaluating LV GLS can assist in risk stratification during the critical early period following HT, complementing traditional monitoring methods such as right-heart catheterization and EMB. It is worth noting that some HT patients may exhibit reduced LV GLS despite having a normal or slightly reduced LV EF [17].

The left atrium (LA) and right atrium (RA) have crucial roles in regulating ventricular filling via their reservoir, conduit, and contractile functions in cardiac physiology. In HT recipients, atrial function is influenced not only by ventricular dysfunction but also by surgical factors. In the standard technique, both atria are enlarged as they are connected in the middle portion using a combination of recipient and donor tissue. With the bicaval method, the RA is reconstructed using donor tissue only, while the LA consists of a mixture of donor and recipient tissue, with the extent of residual recipient tissue varying based on the surgical approach. In most cases, the LA roof, including the pulmonary vein ostia and the tissue in between, remains intact [18]. Comparatively, the atria are typically smaller in size when the bicaval method is used compared with the standard technique [18].

Advanced strain echocardiography can be employed to evaluate atrial function in HT recipients. Bech-Hanssen et al. [19] highlighted a significant reduction in atrial reservoir function among HT recipients, evaluated using speckle tracking. This reduction was attributed to elevated PCWP, enlargement of the LA, and impaired longitudinal RV function. The decline in atrial reservoir function is most noticeable when there are elevated filling pressures. Zhu et al. [20] observed alterations in LA function throughout all phases of the cardiac cycle, independent of the surgical technique employed. Peak LA strain was found to be associated with worse LV systolic function, suggesting the potential importance of LA function in HT patients. LA strain measurements are generated from six atrial segments, acquired from both the apical four-chamber and two-chamber views, as shown in Figure 1.

## 4. Assessment of Right Ventricle and Tricuspid Valve

The survival of heart transplantation recipients is significantly influenced by the function of the RV, underscoring the critical clinical significance of thorough assessment.

Early RV dilation is frequently observed in HT patients but tends to resolve within a few weeks [21]. However, persistent RV failure is a well-recognized cause of mortality [22]. Therefore, even when RV systolic function appears to be within the normal range, more subtle RV dysfunction can be discerned with RV strain analysis, providing insights into overall RV performance [22]. The role of RV strain becomes particularly noteworthy when combined with LV-GLS, LV-Twist, and troponin, yielding an area under the curve of 0.89 (with a confidence interval of 0.81–0.93) for the detection of ACR, emphasizing the significant contribution of RV in this diagnostic context [23].

A recent study involving the simple technique of short-axis fractional area change (SAXFAC) suggests a potential link between lower SAXFAC values and significant allograft rejection, with minimal inter-observer discrepancies [24]. However, due to the small sample and infrequent rejection cases, these findings should be considered preliminary.

While mitral and aortic valve abnormalities are not commonly encountered in HT patients, tricuspid regurgitation (TR) is quite prevalent with reported occurrence ranging widely, from 19% to 84% [25,26]. This variability hinges on factors like the definition of significant TR, the timing of diagnosis, and the surgical approach employed during transplantation. Alongside this, other types of valvular dysfunction are also documented, including pulmonary insufficiency at 42% and mitral insufficiency at 32% [26]. In the early phases after HT, TR can be attributed to elevated pulmonary artery pressure and resistance in the recipient, and it usually resolves within a year following the surgery [25]. However, if TR persists beyond this period, it is associated with RV failure and increased mortality [25].

The geometric changes in the atria after heart transplantation underlie the predominance of atrioventricular insufficiencies among HT patients [27]. The genesis of TR post-HT is attributed to two principal mechanisms: functional and anatomic [28]. Geometric distortion of the tricuspid annulus is the fundamental driver behind functional tricuspid regurgitation, with several risk factors contributing, such as the utilization of a biatrial technique for right atrial anastomosis, elevated preoperative pulmonary vascular resistance in the recipient, and donor–recipient mismatches. Notably, adopting the bicaval technique has substantially curtailed the incidence of functional tricuspid insufficiency in HT recipients [29]. Anatomic TR, on the other hand, primarily stems from damage to the chordae tendineae due to EMBs conducted when GR is suspected [30]. It is worth noting that the presence of TR in HT recipients is correlated with an increased risk of mortality and a progressive decline in renal function [31].

While transoesophageal echocardiography (TEE) has a well-defined role during intra-operatory management [32,33], it has limited application in HT patients’ follow-up, being its indications mainly related to valvular abnormalities and cardiac masses such as intra-cavitary thrombi or tumors [34,35]. Notably, ETE in HT patient could be technically challenging, requiring non-standard imaging planes and major adjustments in the position of the probe or transducer angle, in the setting of significant anatomic variability due to the different orientation of the transplanted heart in the chest [34].

## 5. Stress Echocardiography

Stress echocardiography (SE) is a commonly used functional imaging test in which the most frequently used stressor is dobutamine [36]. However, due to the diminished heart-rate response to exercise resulting from the cardiac denervation state in HT patients, exercise protocols have limited sensitivity (15–33%) [36]. Several factors support the role of SE in the diagnosis of inducible ischemia in HT patients. Coronary artery disease (CAD) in these patients is often diffuse, making the myocardium more vulnerable to demand ischemia. Additionally, there is impaired coronary collateral circulation. From a technical perspective, patients with cardiac denervation may exhibit an exaggerated chronotropic response to dobutamine infusion [37].

Numerous studies have evaluated the ability of dobutamine SE (DSE) to detect inducible ischemia [36,37,38,39,40]. The sensitivity, specificity, positive predictive value (PPV), and negative predictive value (NPV) of DSE in diagnosing CAV range from 63% to 95%, 55% to 95%, 40% to 92%, and 62% to 92%, respectively. These values vary depending on the angiographic definitions of CAV used, which can range from any angiographic abnormality to significant stenosis [37,38,39,40]. Data regarding the sensitivity and NPV of DSE to detect any stage of CAV are inconsistent, with a study by Eroglu et al. showing a relatively modest sensitivity value of 63% [40]. These discrepancies may be explained by the timing of DSE after HT. DSE is not recommended for routine surveillance of CAV beyond 5 years post-HT due to the high prevalence of the disease and the low NPV of DSE [41]. Furthermore, DSE results do not provide additional prognostic utility in HT patients.

The integration of novel echocardiographic parameters, such as strain imaging, with DSE has shown to be more accurate than visual assessment alone in diagnosing CAD in non-transplant populations [41,42]. Further studies utilizing these techniques may potentially enhance the sensitivity of DSE in detecting early-stage CAV and provide incremental prognostic value. The evaluation of coronary flow reserve (CFR) to assess the presence of microvascular disease is not routinely performed in HT patients, despite encouraging evidence from previous studies [43,44]. Reduced CFR has also been shown to be associated with major cardiovascular events in HT patients [43]. In fact, HT patients with normal systolic function and no evidence of CAV may still exhibit coronary microvascular impairment due to structural remodeling and loss of vasodilatory capacity in the microvasculature [44].

## 6. The Role of Coronary Computed Tomography Angiography and Nuclear Imaging in the Follow-Up of Heart Transplant Recipients

CCTA is emerging as a non-invasive diagnostic tool in the follow-up of HT recipients, offering high sensitivity and negative predictive values in detecting CAD [45]. Several studies have demonstrated that CCTA has lower costs, shorter examination time, reduced radiation exposure, and lower risk of vascular complications compared with ICA [46]. Additionally, last-generation CCTA machines have good accuracy in analyzing more distal coronary segments compared with ICA [46]. However, achieving a low heart rate is necessary to obtain optimal diagnostic accuracy with CCTA. Guidelines recommend a resting heart rate of less than 60 bpm for ideal image quality [5]. HT patients often have an elevated resting heart rate due to cardiac denervation, resulting in the loss of vagal rate regulation. Dual-source CT (DSCT) or multisegment reconstruction (MSR) techniques enable good image quality at low radiation dosages, even at higher heart rates (>75 bpm) [47]. Interestingly, one study showed that CCTA images acquired in HT patients at an average heart rate of 74 bpm were still superior to those obtained in non-HT patients at 73 bpm. This is attributed to the denervation of the transplanted heart, which leads to an absence of heart-rate variation. Further studies have confirmed that a regular heart rate is associated with better image quality [48,49].

Advancements in CCTA imaging, such as fractional flow reserve derived from CT (FFR-CT), have been studied for the HT population. CT-derived FFR can identify early CAV and analyze all coronary artery branches [50]. FFR-CT evaluations have demonstrated the ability to predict mortality and the risk of re-transplantation [51]. The use of CCTA in the evaluation of CAV has been included in the guidelines for the management of HT patients as a Class IIb recommendation (Level of Evidence: C) [52]. However, limited data are available on the role of CCTA in the follow-up of HT patients. One study conducted by Rohnean et al. reported slow progression of CAV over a 5-year follow-up period as detected with qualitative CCTA assessment [53]. The time to significant stenosis was longer than 3 years among the 62 HT patients evaluated, leading to a recommendation of a 2-year interval between follow-up studies in patients with a normal baseline CT. The anatomical and histopathological characteristics of CAV differ from those of atherosclerotic disease. CAV typically affects small, distal vessels, while the luminal narrowing of larger epicardial coronaries only occurs in advanced stages. Unlike the eccentric and focal lesions observed in atherosclerotic disease, CAV manifests as concentric intimal hyperplasia without positive wall remodeling, making it easily overlooked by an inexperienced examiner [54].

The integration of intravascular ultrasound (IVUS) with ICA reveals approximately 19% more cases of CAV, making it the most sensitive test available for monitoring CAV progression in the clinical setting [55]. However, CCTA has been proposed as an alternative to IVUS for routine follow-up of HT patients due to its superior spatial resolution [56]. In a recent meta-analysis, the sensitivity, specificity, positive predictive value, and negative predictive value of CCTA for detecting CAV were reported as 97%, 81%, 78%, and 97%, respectively [57]. Nevertheless, there is currently no standard method available for the assessment of CAV on CCTA images, and distinguishing CAV from atherosclerotic lesions can be challenging (examples in Figure 2 and Figure 3).

The visual assessment of coronary lesions in CCTA can be subjective and dependent on the reader’s experience [58]. To address this limitation, quantitative software tools have been introduced for the assessment of stenosis severity in CCTA. These tools provide more objective measurements and help evaluate the progression of coronary vessel wall volume over time.

A study by Karolyi et al. demonstrated progression of coronary vessel wall volume within the first two years after HT using quantitative CCTA analysis [59]. This progression was primarily due to non-calcified lesions that caused mild luminal narrowing. When assessing the characteristics of these lesions, the main components of non-calcified lesions showed high attenuation (131–350 HU), corresponding to fibrous tissue. A smaller percentage of the lesions exhibited intermediate attenuation (75–130 HU), consistent with fibro-fatty tissue, and a low-attenuation (<75 HU) component, indicative of lipid-rich content [60]. These findings are characteristic of CAV, where coronary lesions tend to be diffuse, and significant focal luminal narrowing develops in only a small number of patients.

Positron emission tomography (PET) myocardial perfusion imaging has gained attention in this regard, with multiple studies in the past decade demonstrating its value in quantifying myocardial blood flow for diagnosing, prognosticating, and monitoring CAV in HT recipients [61,62,63,64]. PET scans with myocardial blood flow (MBF) quantification are being explored as a non-invasive alternative to ICA. A study by Abadie et al. [65] aimed to validate the diagnostic and prognostic value of a PET/CT-based algorithm for diagnosing CAV in a large population of HT patients. The algorithm exhibited strong negative predictive value for moderate-to-severe CAV and positive predictive value for significant CAV. Patients with higher PET CAV scores showed increased risk of adverse events. In another recent study, the utility of single-photon emission computed tomography (SPECT) was explored to detect CAV [66]. The study findings revealed a robust concordance between SPECT and PET scans when evaluating critical factors such as MBF and myocardial flow reserve (MFR). This observation implies that SPECT exhibits promise in identifying cases of moderate-to-severe CAV. Nevertheless, to firmly establish these results, additional extensive research involving larger cohorts is mandatory.

## 7. Diagnostic Challenges and Potential Benefits of CMR in the Follow-Up of Heart Transplant Recipients

CMR is indeed considered the gold-standard imaging modality for assessing cardiac morphology, ventricular volumes, systolic function, and myocardial mass [67]. It offers comprehensive evaluation of the heart, allowing for the assessment of various parameters and functions.

In the context of HT recipients, CMR has shown utility in assessing the activity of inflammatory changes in the myocardium. This includes the detection of myocardial edema, hyperemia, capillary leak, and irreversible injury using a combination of non-contrast techniques such as T2-weighted imaging, as well as parametric mapping techniques such as T1 and T2 mapping. These techniques provide valuable information about the presence and extent of inflammatory processes within the transplanted heart [68].

The study by Vermes et al. [69] supports the use of a multi-parametric sequential approach using CMR to diagnose acute rejection in HT recipients. They found that combining basal T2 mapping with basal extracellular volume (ECV) measurement yielded the best diagnostic accuracy. This approach has the potential to reduce the need for invasive EMBs by more than 50%. Interestingly, the study also found that T1-mapping values before and after contrast injection did not show significant increases in the presence of rejection. This highlights the limitations of measuring absolute myocardial T1 values, as they can be influenced by various factors, such as the acquisition scheme, magnetization transfer, flow, and T2 effects [70].

Overall, CMR, with its multi-parametric approach, offers valuable insights into the inflammatory processes occurring in HT recipients and can help guide clinical decision making and reduce the need for invasive procedures like EMBs.

The presence of an unexplained fall in LVEF combined with a non-diagnostic EMB poses a diagnostic dilemma in patients several years post-cardiac transplantation. The limitations of EMB in this group include hindrance by endomyocardial fibrosis from prior biopsy sites and difficulties in central venous access due to multiple prior cannulations [71]. Additionally, diffuse myocardial fibrosis, which is common in long-term HT recipients, can make it difficult to obtain adequate myocardial samples for biopsy [72]. Therefore, the development of a non-invasive test for GR is particularly valuable in these patients.

CMR offers high spatial resolution and accurate assessment of ventricular volume and function using cine imaging. While nuclear cardiac gated blood pool imaging can also provide LVEF measurements, CMR’s advantage lies in the absence of ionizing radiation, which is beneficial for patients on long-term immunosuppressive therapy [71]. Furthermore, CMR allows for the calculation of GLS, which may serve as an early-integrated biomarker of pathologies affecting the subendocardium, such as CAV and allograft failure [72]. Moreover, abnormal GLS has been associated with an increased risk of adverse cardiac events in HT patients [72,73].

Studies such as the randomized trial conducted by Anthony et al. [74] have demonstrated the feasibility of CMR-based surveillance for GR, reducing the potential complications associated with EMB-based surveillance. In the pediatric population, where signs of allograft rejection may be more difficult to appreciate, CMR can play an important role in cardiac rejection surveillance. EMB in children often requires general anesthesia, adding to the procedural risk and invasiveness [75,76,77]. CMR offers non-invasive myocardial tissue characterization and can be informative in the pediatric setting, providing a surveillance protocol that requires no intravenous cannulation, gadolinium administration, or breath holding during T2-mapping sequences [74].

The use of CMR-derived GLS, combined with data from cine imaging, perfusion imaging, late gadolinium enhancement (LGE), T1 mapping, and T2 mapping, holds promise in improving the prognostic assessment of cardiac allograft rejection [72]. However, prospective studies are needed to determine whether a GLS-guided strategy is associated with improved long-term outcomes. Additionally, in HT recipients with chronic kidney disease, CMR-FT GLS can provide prognostic information without the need for contrast administration, addressing concerns about the risk of nephrogenic systemic fibrosis associated with gadolinium-based contrast agents [72].

In the assessment of valvulopathies in HT patients, cardiac MRI is frequently employed as a secondary evaluation following the standard ultrasound examination [73]. Cardiac MRI is particularly valuable for patients with an inadequate acoustic window. Sometimes, the way the transplanted heart is placed in the chest can make it difficult to get clear heart ultrasound images. This can also happen if a person gains weight due to the immunosuppressive medications [78].

## 8. Conclusions

New non-invasive techniques have been proposed and tested to reduce the need for invasive examinations in the follow-up of HT patients.

Advanced echocardiography techniques, such as strain imaging, and tissue Doppler imaging, offer enhanced assessment of cardiac function and can help identify abnormalities in HT patients. GLS, in particular, has shown promise as an early marker of cardiac allograft rejection [72].

CMR is considered the gold-standard imaging modality for assessing cardiac morphology, ventricular volumes, systolic function, and myocardial mass in HT patients [67]. CMR also allows for the evaluation of myocardial tissue characteristics using various sequences, such as T1 and T2 mapping, LGE, and parametric mapping techniques [68]. These techniques can help detect inflammation, myocardial edema, fibrosis, and irreversible injury, providing valuable insights into graft health and the presence of complications.

Furthermore, CCTA is particularly useful in evaluating CAV in HT recipients. It allows for the non-invasive assessment of coronary artery stenosis, plaque burden, and morphology. Quantitative software tools have been developed to assess stenosis severity, aiding in the detection of CAV progression and potentially reducing the need for ICA [59].

Recent research has underscored the promise of nuclear myocardial perfusion imaging involving the quantification of myocardial blood flow as a non-invasive approach to diagnosing and predicting CAV. Additionally, the agreement between SPECT and PET scans in evaluating essential factors implies the potential of SPECT for identifying moderate-to-severe CAV cases.

By combining information, clinicians can obtain a comprehensive evaluation of HT patients, identifying complications such as rejection and CAV. This multimodality approach offers the potential to optimize patient care by providing detailed information without the need for invasive procedures, reducing patient discomfort and potential complications.

## 9. Future Directions

Highlighting the changing landscape of post-HT patient care, the importance of progress in non-invasive and contrast-saving diagnostic tools becomes paramount.

HT remains the gold-standard treatment for patients with advanced heart failure, and the use of these advanced imaging modalities can greatly impact the management and outcomes of these patients.

Chronic inflammation plays a significant role in the development of complications after HT, including graft rejection and CAV. The impact of chronic inflammation on epicardial and pericoronary fat is an area that requires further investigation with the use of CCTA. In addition, CMR multi-parametric mapping techniques, such as T1 and T2 mapping, offer the potential to assess inflammation and tissue characterization in a non-invasive manner. These mapping techniques may become a first-line strategy in the surveillance of rejection, providing valuable information about the myocardium and aiding in the early detection of rejection episodes.

In addition to rejection, microvascular disease can contribute to cardiac dysfunction in HT patients. Stress CMR and nuclear imaging can be useful in detecting microvascular disease even in the absence of coronary artery stenosis and help identify patients who may benefit from targeted therapies.

To optimize the care of HT patients, it is crucial to have specialized centers with dedicated HT teams and expertise in multimodality imaging. Large HT centers with experience in various imaging modalities can provide comprehensive evaluation and management. This multidisciplinary approach ensures that patients receive the best possible care, incorporating the advantages of non-invasive imaging techniques and tailoring treatment strategies based on individual patient needs.

While advancements in non-invasive imaging techniques have shown great promise, further studies are necessary to strengthen the evidence base and establish their role in routine clinical practice. These studies could add significant evidence to ascertain the advantages of these modalities in areas such as cost effectiveness, time efficiency, and the early detection of complications. By incorporating these tools into the follow-up care of HT patients, the healthcare system can potentially benefit from improved patient outcomes and resource allocation.

## Figures and Tables

**Figure 1 diagnostics-13-02818-f001:**
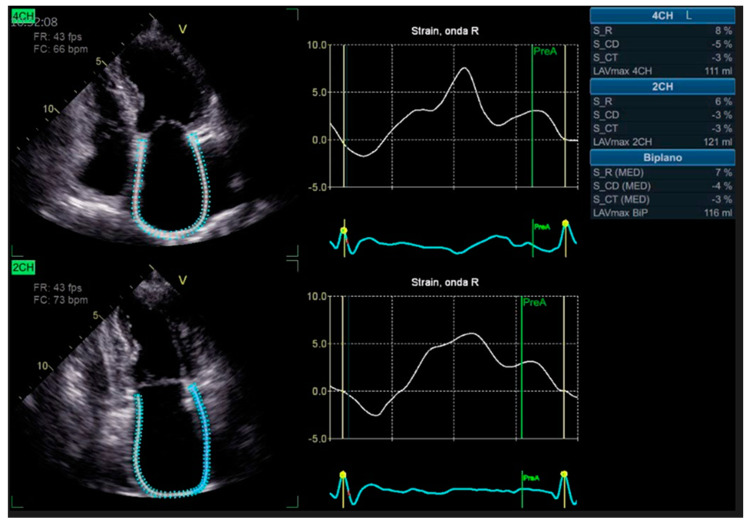
Two-dimensional left atrial strain using speckle tracking echocardiography obtained from the apical four-chamber (**upper** panel) and two-chamber (**lower** panel) views.

**Figure 2 diagnostics-13-02818-f002:**
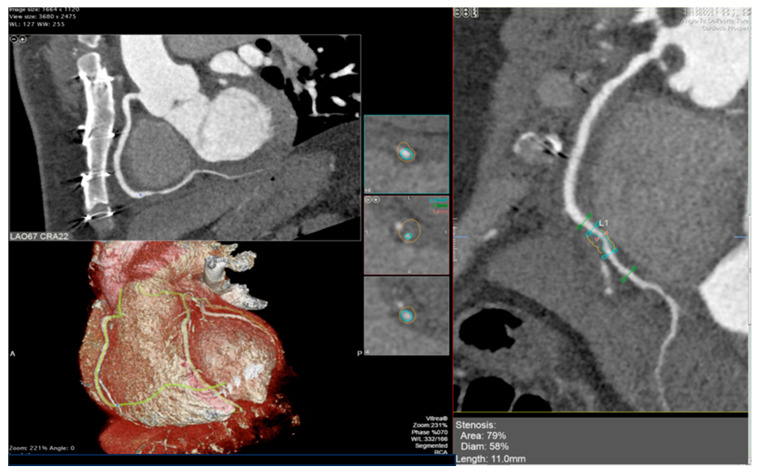
Cardiac computed tomography images showing cardiac allograft vasculopathy grade III according to the International Society for Heart and Lung Classification. Right coronary artery reconstruction with a calcific coronary plaque at the second tract assessed with cardiac computed tomography angiography (**right** panel) and his orthogonal views (**left** panel).

**Figure 3 diagnostics-13-02818-f003:**
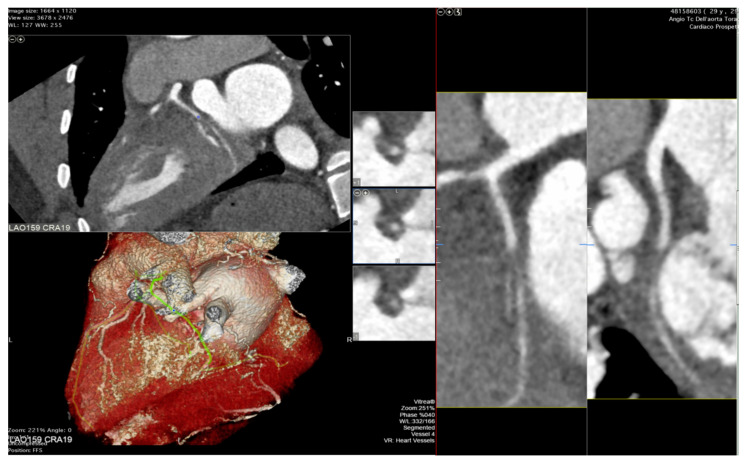
Cardiac computed tomography images showing cardiac allograft vasculopathy grade IV-A according to the International Society for Heart and Lung Classification with a non-calcific, sub occlusive coronary plaque at the second tract of the circumflex coronary artery assessed with cardiac tomography angiography.

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
