# Peer review of "New Non-Invasive Imaging Technologies in Cardiac Transplant Follow-Up: Acquired Evidence and Future Options"

_diagnostics, 2023, doi:10.3390/diagnostics13172818_

Round 1

Reviewer 1 Report

1. Author should properly cite all sources in the paper; for instance, citations 2, 3, and 6 are not included.

2. Can the author clarify whether CMR is cardiac or cardiovascular magnetic resonance imaging.

3. When an abbreviation is used for the first time in a work, the author should extend it. For instance, ISHLT in line 66 and ESC in line 76.

4. Can the author reexamine the cardiac computed tomography angiography? For example, in line 31, is the term "Coronary Computed Tomography Angiography" (CCTA).

5. can the author add citations for figures 2 and 3.

6. Can the author elaborate on "these non-invasive techniques" on line 27.

7. Can the author explain the numerous studies in line 200 and correctly cite the source from which the data were extracted.

8. Overall, the author should enhance the manuscript by adding more references and providing a deeper explanation of the subsections.

 Extensive editing of English language required

Author Response

Reviewer #1.

Thank you for bringing your annotations to our attention. We sincerely appreciate your diligence in reviewing our paper and identifying areas for improvement. Your feedback is invaluable in maintaining the accuracy and credibility of our work. We are grateful for your assistance in making our paper more robust.

1. Author should properly cite all sources in the paper; for instance, citations 2, 3, and 6 are not included.

R: We apologize for the oversight in citing sources 2, 3, and 6, and we rectified this by ensuring proper citation in the paper.

2. Can the author clarify whether CMR is cardiac or cardiovascular magnetic resonance imaging.

R: We thank the Reviewer for the annotation. We have clarified that CMR is cardiac magnetic resonance imaging.

3. When an abbreviation is used for the first time in a work, the author should extend it. For instance, ISHLT in line 66 and ESC in line 76.

R: We thank the Reviewer for the comment. We have extended all the abbreviations as indicated.

4. Can the author reexamine the cardiac computed tomography angiography? For example, in line 31, is the term "Coronary Computed Tomography Angiography" (CCTA).

R: Thank you for your feedback and attention to detail. We have carefully reviewed the terminology and will ensure its accuracy in the revised version of the manuscript.

5. Can the author add citations for figures 2 and 3.

R: Thank you for your attention to the references for figures 2 and 3. We would like to clarify that the images we have included in the paper are derived from our own patient cases, and as such, we didn't include specific references. 

6. Can the author elaborate on "these non-invasive techniques" on line 27.

R: We thank the Reviewer for the comment. According to the suggestion, we have specifically outlined the non-invasive techniques being referred to. These encompass advanced imaging methods such as PET myocardial perfusion imaging, strain imaging, tissue Doppler imaging in echocardiography, multiparametric mapping techniques like T1 and T2 mapping in cardiac magnetic resonance (CMR), and cardiac computed tomography (CCT). These techniques collectively aim to provide alternatives to invasive procedures, contributing to improved patient care and reduced reliance on more risky interventions.

7. Can the author explain the numerous studies in line 200 and correctly cite the source from which the data were extracted.

R: We appreciate your feedback. The numerous studies mentioned in line 200 have been elaborated upon (references 37-41), and the relevant sources from which the data were extracted have been properly cited in the references. 

8. Overall, the author should enhance the manuscript by adding more references and providing a deeper explanation of the subsections.

R: Thank you for your stimulating feedback. The manuscript has been enhanced by incorporating additional references to support the presented concepts and by providing more in-depth explanations in the various subsections. This inclusion of additional references and detailed explanations contributes to a more comprehensive and well-supported presentation of the discussed topics.

======================================

Reviewer 2 Report

I suggest authors revise their conclusion, it should be shortened.

Please, add about TOE and 24h ECG monitoring.

I suggest authors check their manuscript and switch words into British English.

Author Response

Reviewer #2.

We acknowledge your observations regarding our manuscript, and we recognize the value of your review in identifying areas for improvement. Your feedback serves as a valuable resource in ensuring the accuracy and quality of our work. Your contributions to enhancing the strength of our paper are duly noted and appreciated.

1. The abstract is poorly written and needs improvement. I suggest authors revise their conclusion, it should be shortened.

R: Thank you for your valuable feedback. We appreciate your insights and have taken your suggestions into consideration. Our abstract has been revised to be more impactful. We are committed to enhancing the quality of our work and ensuring that also our conclusions are well-structured and aligned with the content of the paper. We believe that the revised version now better encapsulates the key messages of our review.

2. Please, add about TOE and 24h ECG monitoring.

R: In response to your valuable input, we have thoughtfully integrated relevant information about TOE and expounded upon its application within the context of heart transplant patients. While TOE itself lacks specific indications for HT recipients, we have taken care to incorporate relevant commentary within the section addressing valvular abnormalities, accommodating any potential relevance. It is important to note that we have excluded the discussion on 24-hour ECG monitoring, as it lies outside the purview of the manuscript's focus, which primarily centers around cardiac imaging. To provide utmost clarity, we have further refined the manuscript's title by including the term "imaging," clarifying that our discourse pertains exclusively to this aspect of HT follow-up. Your kindest guidance has undeniably contributed to the meticulous refinement of our paper.

======================================

Reviewer 3 Report

Recent acquisitions, progress, and new technologies have significantly improved noninvasive techniques in their capacity to investigate heart and vessels thoroughly. In patients who have undergone a heart transplant, the possibility of carrying out follow-up reducing invasive procedures (cardiac catheterization, angiography, endomyocardial biopsy) is of significant value from a clinical and quality-of-life point of view.  This study interestingly brings up the evidence showing promising data which will impact guidelines and clinical practice even if some studies are still needed. 

Need some minor editing of english.

Author Response

  1. Recent acquisitions, progress, and new technologies have significantly improved noninvasive techniques in their capacity to investigate heart and vessels thoroughly. In patients who have undergone a heart transplant, the possibility of carrying out follow-up reducing invasive procedures (cardiac catheterization, angiography, endomyocardial biopsy) is of significant value from a clinical and quality-of-life point of view.  This study interestingly brings up the evidence showing promising data which will impact guidelines and clinical practice even if some studies are still needed.

R: A heartfelt thank you for your insightful comment. Your observation resonates deeply with the essence of our study. Indeed, recent advancements and the emergence of novel technologies have revolutionized noninvasive techniques, greatly enhancing their ability to comprehensively explore the heart and vessels. The potential to conduct follow-up assessments in heart transplant patients while reducing the need for invasive procedures such as cardiac catheterization, angiography, and endomyocardial biopsy holds profound clinical and quality-of-life implications. We are genuinely appreciative of your acknowledgment of our study's significance, which sheds light on promising findings that stand to influence guidelines and clinical practices, despite the ongoing need for further studies. Your thoughtful input is highly valuable to us.

======================================

Round 2

Reviewer 1 Report

The manuscript titled “New Non-invasive Imaging Technologies in Cardiac Transplant Follow-up: Acquired Evidence and Future Options” is well-written and addresses all initial reviewer comments in an effective manner. The authors have made substantial revisions to the manuscript in response to the feedback, resulting in a presentation that is more clear and coherent. The revisions have greatly improved the overall quality of the paper.

Comments:

1.     The author specified that Figure 2 and Figure 3  were obtained from their own patients (was informed consent obtained from the patients? ), but for ethical permission, the author stated that ethical review board approval was not required. Can the author clarify?

Author Response

Thank you for your valuable comment and inquiry regarding the ethical aspects of the study. We appreciate your attention to this important matter. We would like to clarify that our study adhered to ethical guidelines and patient consent protocols. All patients provided informed consent, and we followed a standardized consent form approved by our institutional review board. This form (see the blank form attached) ensured that patients were well-informed about the nature of their participation and the usage of their data for research purposes. Therefore we added in the section Institutional Review Board Statement that "Consent for the publication of pictures was obtained from individuals included in our study".

If you have any further questions or concerns, please feel free to reach out. Your feedback is greatly valued and helps us enhance the transparency and ethical rigor of our research endeavors.